Identification and molecular characterization of mutations in nucleocapsid phosphoprotein of SARS-CoV-2

http://orcid.org/0000-0001-5478-526X Azad Gajendra Kumar gkazad@patnauniversity.ac.in
Department of Zoology, Patna University , Patna, Bihar , India
Newsome Timothy
Electronic publication date: 2021 Jan 4
Publication date: 2021
Volume: 9
Electronic Location ID: e10666
Received 2020 Jun 28; Accepted 2020 Dec 8
Copyright: © 2021 Azad
Copyright year: 2021
Copyright holder: Azad
License: This is an open access article distributed under the terms of the Creative Commons Attribution License, which permits unrestricted use, distribution, reproduction and adaptation in any medium and for any purpose provided that it is properly attributed. For attribution, the original author(s), title, publication source (PeerJ) and either DOI or URL of the article must be cited.
License URL: https://creativecommons.org/licenses/by/4.0/

Keywords: SARS-CoV-2, COVID-19, Nucleocapsid Phosphoprotein (N protein), USA, Mutations, Infectious disease

Funding: The author received no funding for this work. The funders had no role in study design, data collection and analysis, decision to publish, or preparation of the manuscript The author received no funding for this work.

==============================
SARS-CoV-2 genome encodes four structural proteins that include the spike glycoprotein, membrane protein, envelope protein and nucleocapsid phosphoprotein (N-protein). The N-protein interacts with viral genomic RNA and helps in packaging. As SARS-CoV-2 spread to almost all countries worldwide within 2–3 months, it also acquired mutations in its RNA genome. Therefore, this study was conducted with an aim to identify the variations present in N-protein of SARS-CoV-2. Here, we analysed 4,163 reported sequence of N-protein from United States of America (USA) and compared them with the first reported sequence from Wuhan, China. Our study identified 107 mutations that reside all over the N-protein. Further, we show the high rate of mutations in intrinsically disordered regions (IDRs) of N-protein. Our study show 45% residues of IDR2 harbour mutations. The RNA-binding domain (RBD) and dimerization domain of N-protein also have mutations at key residues. We further measured the effect of these mutations on N-protein stability and dynamicity and our data reveals that multiple mutations can cause considerable alterations. Altogether, our data strongly suggests that N-protein is one of the mutational hotspot proteins of SARS-CoV-2 that is changing rapidly and these mutations can potentially interferes with various aspects of N-protein functions including its interaction with RNA, oligomerization and signalling events.

Introduction

In the late December 2019, Wuhan, the Hubei province of China, reported a surge in hospitalisation due to pneumonia-like symptoms (Zhu et al., 2020). The causative agent was identified as a severe acute respiratory syndrome coronavirus 2 (SARS-CoV-2) that shares close similarity with earlier known SARS-CoV (Chen et al., 2020). The SARS-CoV-2 is highly contagious, which led to its rapid spread worldwide, and in March 2020, the World Health Organization (WHO) declared the outbreak a pandemic. The disease caused by SARS-CoV-2 has been named as coronavirus disease 19 (COVID-19), and exhibits mild to severe respiratory distress in the infected individuals. As of 28 June 2020, the COVID-19 has affected all countries worldwide with close to 10 million reported cases and 0.5 million confirmed deaths. Further, the epidemiological studies revealed that the mortality rate from COVID-19 is significantly higher among individuals over 60 years of age with weak immunity (Liu et al., 2020).

The SARS-CoV-2 has positive sense, single stranded RNA genome of approximately 29.8 kb (Wu et al., 2020b). The majority of viral genome encodes non-structural proteins that are proteolytically processed from a single Orf1ab polypeptide. SARS-CoV-2 genome also encode four structural proteins, including the spike glycoprotein (S), membrane protein (M), envelope protein (E) and nucleocapsid phosphoprotein (N) (Wu et al., 2020a). The S, M and E proteins are located in the lipid bilayer of the virus and contribute to the formation of viral envelope; however, the N-protein contributes to the viral genomic RNA packaging and remains embedded in the central core of the virion. N-protein binds with viral genomic RNA and forms helical structure to maintain the structural integrity of RNA genome (Chang et al., 2014). This is one of the most abundant structural proteins encoded by the SARS-CoV-2 genome. The SARS-CoV-2 N-protein resembles N-protein from other RNA viruses, known to modulate host intracellular machinery and also involved in the regulation of virus life cycle (McBride, Van Zyl & Fielding, 2014). Evidence show that N-protein is recruited to the Replication-Transcription Complexes (RTC) via Nsp3 and plays a crucial role in coronaviral life cycle (Cong et al., 2019). The abrogation of this interaction impairs the stimulation of genomic RNA and viral mRNA transcription in vivo and in vitro. Furthermore, the N-protein interactions with M promotes completion of viral assembly by stabilizing N protein-RNA complex, inside the internal virion (Astuti & Ysrafil, 2020).

The crystal structure of N-protein revealed two distinct domains at N and C terminus (Kang et al., 2020). The domain present towards the N terminus is also known and RNA-binding domain (RBD). The C terminal side harbours dimerization domain which interacts with other N-protein to make dimer. Apart from these two domains there are three intrinsically disordered regions (IDRs) at N and C terminal ends as well as between the RBD and dimerization domain of N-protein. Since, this protein plays critical role in packaging of SARS-CoV-2 RNA genome, the mutations in N-protein or interfering its function can lead to diverse outcome on viral life cycle (Rabi Ann Musah, 2005; Chenavas et al., 2013).

Moreover, the study of N-protein is also important because of its unique immunological properties. For instance, earlier study with SARS N-protein has shown that this protein is a potential candidate for vaccine development because it can induce a strong immunological response (Liu et al., 2006). A recent study revealed that the B and T cell epitopes of N-protein of SARS-CoV-2 shows close resemblance with that of SARS-CoV indicating that immune targeting of these identical epitopes may offer protection against this virus (Ahmed, Quadeer & McKay, 2020). Moreover, the sera of COVID-19 patients contains abundant amount of IgA, IgM and IgG antibodies against N-protein antigen demonstrating the importance of this antigen in host immunity and diagnostics (Shang et al., 2005; Zeng et al., 2020). Therefore, the N-protein is one of the candidate target molecule that needs to be properly studied to understand its role in virus pathogenesis, vaccine development and pharmacological implications. Here, we compared the N-protein sequences obtained from USA with first reported sequence from China to identify the variations present between them. We have identified 107 mutations and their impact on N-protein structure and function are discussed.

Materials and Methods

Sequence retrieval from NCBI-virus-database

The NCBI-virus-database stores the deposited sequences of SARS-CoV-2 which is updated regularly as the new sequences are reported. As of 23 June 2020, 4,163 SARS-CoV-2 sequences of N-protein were deposited from USA. We downloaded these sequences and used them for analysis in this study. The first reported N-protein sequence from Wuhan was used as reference sequence or wild type sequence (Wu et al., 2020b). The protein accession identification number of reference sequence used in this study is YP_009724397 and rest of the 4,163 IDs (reported from USA) are mentioned in Table S1.

Multiple sequence alignment by Clustal-Omega program

To identify the mutations present in the SARS-CoV-2 N-protein reported from USA, we did multiple sequence alignments and compared them with the first reported N-protein sequence (YP_009724397) from Wuhan, China as described earlier (Azad, 2020). The multiple sequence alignment was performed using Clustal Omega tool (Madeira et al., 2019).

Calculation of free energy and vibrational entropy between wild type and mutant N-proteins

In order to measure the impact of mutations identified in this study on the structural dynamicity and stability of N-protein, we calculated the differences in free energy (ΔΔG) and vibrational entropy (ΔΔSvib) ENCoM between wild type and mutants as described earlier (Chand, Banerjee & Azad, 2020a). This analysis was performed by DynaMut program (Rodrigues, Pires & Ascher, 2018). To perform DynaMut protein modelling we used RCSB protein ID: 6VYO (Kang et al., 2020) for RBD molecular modelling and RCSB protein ID: 6WJI for dimerization domain molecular modelling of N-protein. DynaMut also provide the visual representation of fluctuation in protein structure. The blue colour represents gain in rigidity and red colour represents gain in flexibility upon mutation.

Structural representation of N-protein domains

The UCSF Chimera program (Pettersen et al., 2004) was used for the interactive visualization and analysis of molecular structures and related data. High-quality images were generated as output file from this program. For structural representation, RCSB protein ID: 6VYO and 6WJI was used for RNA-binding domain and dimerization domain of N-protein respectively.

Generation of weblogo to show conservation of N-protein sequences

The weblogo was generated using a webserver as described earlier (Crooks et al., 2004). The overall height of the stack indicates the sequence conservation at that position. For this analysis, all N-protein sequences (4,163) reported from USA and the reference sequence (YP_009724397) was used. The sequence logo was generated by multiple sequence alignment of these N-protein sequences.

Results

Identification of mutations in IDR1, IDR2 and IDR3 of N-protein

The crystal structure of N-protein of SARS-CoV-2 has been recently solved (Kang et al., 2020), the structural details show it is comprised of three distinct regions; the N terminal domain (contains RNA-binding domain), C terminal domain (contains dimerization domain) and IDRs as shown in Fig. 1A. There are three IDRs in N-protein; IDR1 (at the N terminal end), IDR2 (between RBD and CTD) and IDR3 (at the C terminal end). IDR2 is also referred as linker region (LKR) because it connects RBD and dimerization domain of N-protein. In order to identify the variations present in N-protein of SARS-CoV-2 reported from the USA, we performed multiple sequence alignments. Here, we used Clustal Omega program to align 4,163 N-protein polypeptide sequences from USA and compared them with the first reported sequence from Wuhan, China.

Figure 1 The schematic structure of Nucleocapsid Phosphoprotein (N protein) of SARS-CoV-2.

(A) The N-protein is comprised of 419 residues. The RNA-binding domain, dimerization domain, intrinsically disordered regions including IRD1, IRD2 and IRD3 are labelled. (B and C) Cartoon representation of crystal structure of the RNA-binding domain (RCSB protein ID-6VYO) and dimerization domain (RCSB protein ID-6WJI) of N-protein. (B) Demonstrates the RBD while (C) represents dimerization domain of N-protein. The identified mutated amino acids (single letter code) along with its respective position in polypeptide sequence are shown. The structural representations were made using UCSF chimera software tool.

Our analysis identified eighteen mutations in IDR1 (Table 1). The IDR1 is present from 1 to 43 residues towards the N terminal end of N-protein. These eighteen mutations correspond to approximately 40% (18 out of 43) of the residues of IDR1. Among these the most frequently mutated residues are Gly and Arg (both are mutated at four positions) and Pro residue is mutated at three different positions in IDR1 (Table 1).

Table 1 IDR1 mutations.

The location and details of mutations identified in IDR1 of N-protein are shown.

S. No.	Wild type residue	Position of mutation	Mutated residue	
1	Asp	3	Tyr	
2	Asn	4	Asp	
3	Pro	6	Thr	
4	Gln	9	His	
5	Pro	13	Leu	
6	Arg	14	His	
7	Gly	18	Cys	
8	Gly	19	Arg	
9	Pro	20	Leu	
10	Asp	22	Tyr	
11	Ser	23	Thr	
12	Gly	30	Ala	
13	Glu	31	Asp	
14	Arg	32	Leu	
15	Gly	34	Leu	
16	Ala	35	Thr	
17	Arg	36	Leu	
18	Arg	40	Cys	
19	Arg	40	Leu	

Similar analysis with IDR2 identified thirty six mutations which correspond to approximately 45% of residues of IDR2 (Table 2). The IDR2 is present from 181 to 256 residues of the N-protein and connects RBD and dimerization domains. The most frequently mutated residue in IDR2 was found to be Ser, it is mutated at twelve positions. Further, the Ala, Gly and Arg residues are mutated at five positions, respectively.

Table 2 IDR2 mutations.

The location and details of mutations identified in IDR2 are shown.

S. No.	Wild type residue	Position of mutation	Mutated residue	
1	Ser	183	Tyr	
2	Arg	185	Cys	
3	Arg	185	Leu	
4	Ser	187	Leu	
5	Ser	188	Leu	
6	Ser	190	Ile	
7	Arg	191	Leu	
8	Asn	192	Ser	
9	Ser	193	Ile	
10	Ser	194	Leu	
11	Arg	195	Ile	
12	Ser	197	Leu	
13	Pro	199	Ser	
14	Ser	202	Asn	
15	Arg	203	Lys	
16	Arg	203	Met	
17	Gly	204	Arg	
18	Thr	205	Ile	
19	Ala	208	Gly	
20	Arg	209	Lys	
21	Arg	209	Thr	
22	Ala	211	Ser	
23	Gly	212	Cys	
24	Asn	213	Tyr	
25	Gly	215	Ser	
26	Ala	218	Val	
27	Ala	220	Thr	
28	Gln	229	His	
29	Ser	232	Arg	
30	Ser	232	Thr	
31	Met	234	Ile	
32	Ser	235	Pro	
33	Ser	235	Phe	
34	Gly	236	Val	
35	Gly	238	Cys	
36	Gly	243	Cys	
37	Thr	247	Ala	
38	Lys	249	Arg	
39	Ser	250	Phe	
40	Ala	252	Ser	
41	Ser	255	Ala	

Similarly, we identified fifteen mutations in IDR3 (Table 3). The IDR3 is present from 365 to 419 residues towards the C terminal end of N-protein. Most notable mutations are Thr and Ala residues that are mutated at three positions and Pro, Asp and Gln are mutated at two positions, respectively (Table 3). Altogether, we identified sixty nine mutations in intrinsically disordered regions IDR1, IDR2 and IDR3 of N-protein.

Table 3 IDR3 mutations.

The location and details of mutations identified in IDR3 are shown.

S. No.	Wild type residue	Position of mutation	Mutated residue	
1	Pro	365	Ser	
2	Pro	365	Leu	
3	Asp	377	Tyr	
4	Asp	377	Gly	
5	Thr	379	Ile	
6	Gln	380	His	
7	Ala	381	Val	
8	Pro	383	Ser	
9	Pro	383	Leu	
10	Gln	386	Lys	
11	Gln	386	His	
12	Thr	391	Ile	
13	Thr	393	Ile	
14	Ala	397	Ser	
15	Ala	398	Val	
16	Asp	399	Glu	
17	Ser	413	Ile	
18	Ser	416	Leu	

Identification of mutations in RBD and dimerization domain of N-protein

The RBD of N-protein starts from 44th residue till 180th residue. We mapped the mutation in this region of N-protein and our analysis revealed presence of twenty two mutations (Table 4). These twenty two mutations also correspond to approximately 16% of the residues of RBD. Our mutational analysis shows the most frequently mutated residues are Pro and Ala at five positions and Asp at three positions as shown in Table 4.

Table 4 RBD mutations.

The location and details of mutations identified in RBD of N-protein are shown.

S. No.	Wild type
residue	Position of mutation	Mutated residue	
1	Pro	46	Ser	
2	Glu	62	Val	
3	Pro	67	Ser	
4	Asp	81	Tyr	
5	Ala	90	Ser	
6	Ala	119	Ser	
7	Pro	122	Leu	
8	Ala	125	Thr	
9	Asp	128	Tyr	
10	Asn	140	Thr	
11	Pro	142	Ser	
12	Asp	144	Tyr	
13	Asp	144	His	
14	Ile	146	Phe	
15	Pro	151	Leu	
16	Ala	152	Ser	
17	Asn	154	Tyr	
18	Ala	156	Ser	
19	Gln	163	Arg	
20	Thr	166	Ile	
21	Lys	169	Arg	
22	Ser	180	Ile	

Similar analysis with the dimerization domain of N-protein revealed that it harbours sixteen mutations (Table 5). The dimerization domain of N-protein starts from 257th residue till 364th residue. Our mutational analysis shows Thr is mutated at four positions and Asp at three positions. Further, only 14% residues are mutated in this domain which is least among all other regions of the N-protein identified here. Altogether, we identified thirty eight mutations in RBD and dimerization domain of N-protein. We have highlighted the location of amino acids in the representative crystal structure of N-protein that are mutated in RBD (Fig. 1B) and dimerization domain (Fig. 1C)

Table 5 Dimerization domain mutations.

The location and details of mutations identified in the dimerization domain of N-protein are shown.

S. No.	Wild type residue	Position of mutation	Mutated residue	
1	Val	270	Leu	
2	Thr	271	Ile	
3	Gly	284	Glu	
4	Gln	289	His	
5	Ile	292	Thr	
6	Gln	294	Leu	
7	Asp	297	Val	
8	Pro	309	Leu	
9	Met	322	Ile	
10	Ser	327	Leu	
11	Thr	329	Met	
12	Thr	334	Ile	
13	Asp	340	Gly	
14	Asp	340	Asn	
15	Asp	348	Tyr	
16	Thr	362	Ile	
17	Pro	364	Leu	

Subsequently, we also calculated the frequency of each mutation identified in this study. The Table 6 shows the top ten mutants arranged in descending order of their respective frequencies. The R203K mutation is having the highest frequency of 4.9% followed by G204R with 4.7%. Further, we generated weblogo of the 4,163 polypeptide sequences of N-protein to observe their amino acid conservation as shown in Fig. 2. Altogether, we have identified 107 mutations in N-protein that resides in its IDRs and RBD and dimerization domain.

Figure 2 The weblogo diagram showing the conservation status of polypeptide sequence of N-protein.

The sequence weblogo was generated by multiple sequence alignment of 4,164 N-protein sequences. The overall height of the stack indicates the sequence conservation at that position.

Table 6 Frequency of N-protein mutations.

The frequency of top 10 mutations observed in this study.

Mutation	Number of samples that harbour the mutation	% frequency	
R203K	207	4.97357	
G204R	196	4.709274	
E62V	39	0.937049	
A208G	24	0.576646	
S183Y	20	0.480538	
S194L	18	0.432484	
T362I	16	0.384431	
T205I	15	0.360404	
P13L	11	0.264296	
R185C	10	0.240269	

Mutations causes alteration in dynamic stability of N-protein

In order to understand the effect of mutations on the stability of the protein we calculated the differences in free energy (ΔΔG) between wild type and mutants. We performed this analysis using DynaMut program. The positive ΔΔG corresponds to increase in stability while negative ΔΔG corresponds to decrease in stability. We performed this analysis with all of the mutations that reside in RBD and dimerization domain of N-protein. The IDRs do not have proper 3D structure therefore; this analysis is not accurate for those regions. Our data revealed the noticeable increase or decrease in free energy in various mutations as shown in Table 6. The top five positive and negative ΔΔG values are highlighted in Table 6. The maximum increase in ΔΔG was observed for T271I (1.184 kcal/mol) and the maximum negative ΔΔG was obtained for I292T (−1.952 kcal/mol), both of these mutations reside in dimerization domain of N-protein.

We also measured the change in vibrational entropy energy (ΔΔSVibENCoM) between the wild type and the mutants present in RBD and dimerization domain of N-protein (Table 7).Vibration entropy contributes to the configurational-entropy of the proteins (Goethe, Fita & Rubi, 2015). The negative ΔΔSVibENCoM of mutant N-protein corresponds to the increase in rigidification and positive ΔΔSVibENCoM corresponds to gain in flexibility of the protein structure. The maximum positive ΔΔSVibENCoM was obtained for P364L (0.256 kcal.mol−1.K−1) and negative ΔΔSVibENCoM was obtained for G284E (−0.844 kcal.mol−1.K−1). The variation in vibrational entropy between wild type and mutant can also be visualised as shown in Fig. 3. The blue colour corresponds to rigidification in protein structure and red colour corresponds to gain in flexibility upon mutation. The top three positive and negative ΔΔSVibENCoM are shown in Figs. 3A–3F. Altogether, the data obtained from ΔΔG and ΔΔSVibENCoM strongly suggests that the mutations identified in this study can influence N-protein stability and dynamicity.

Figure 3 Visual representation of Δ vibrational entropy energy between wild-type and mutant N protein.

The amino acids residues are colored according to the vibrational entropy change as a consequence of mutation of N-protein. Blue (A–C) represents a rigidification of the structure and red (D–F) represents, a gain in flexibility. (A–C) The top three mutants that show rigidification in structure upon mutation. (D–F) The top three mutants that show gain in flexibility upon mutation. Each panel also shows the mutation and the location of the residues.

Table 7 ΔΔG and ΔΔSvib ENCoM calculations.

S. No.	Mutant	PDB ID	ΔΔG (kcal/mol )	ΔΔSvibENCoM
(kcal.mol-1.K-1)	
1	E62V	6VYO	0.105	0.091	
2	P67S	6VYO	−0.486	0.16	
3	D81Y	6VYO	0.454	−0.425	
4	A90S	6VYO	0.274	0.043	
5	A119S	6VYO	0.073	−0.069	
6	P122L	6VYO	−0.166	−0.049	
7	A125T	6VYO	−0.565	−0.022	
8	D128Y	6VYO	0.846	−0.236	
9	N140T	6VYO	0.318	−0.177	
10	P142S	6VYO	0.26	−0.17	
11	D144Y	6VYO	0.291	−0.293	
12	D144H	6VYO	−0.036	0.06	
13	I146F	6VYO	0.708	−0.837	
14	P151L	6VYO	0.771	−0.14	
15	A152S	6VYO	0.298	−0.051	
16	N154Y	6VYO	−0.096	−0.063	
17	A156S	6VYO	0.428	−0.256	
18	Q163R	6VYO	−0.092	−0.017	
19	T166I	6VYO	0.194	−0.055	
20	K169R	6VYO	0.231	0.077	
21	V270L	6WJI	0.679	−0.194	
22	T271I	6WJI	1.184	−0.472	
23	G284E	6WJI	0.553	−0.844	
24	Q289H	6WJI	0.18	0.181	
25	I292T	6WJI	−1.952	0.186	
26	Q294L	6WJI	0.447	−0.078	
27	D297V	6WJI	−0.113	−0.072	
28	P309L	6WJI	0.887	−0.524	
29	M322I	6WJI	−0.348	0.045	
30	S327L	6WJI	0.894	−0.259	
31	T329M	6WJI	0.569	−0.189	
32	T334I	6WJI	0.236	−0.115	
33	D340G	6WJI	0.398	−0.114	
34	D340N	6WJI	0.194	−0.088	
35	D348Y	6WJI	0.136	−0.121	
36	T362I	6WJI	0.396	0.047	
37	P364L	6WJI	−0.061	0.256	
Note:

The table shows the ΔΔG and ΔΔSvib ENCoM of the mutants present in RBD and dimerization domain of N-protein. DynaMut programe was used to calculate both parameters. The top five positive and negative ΔΔG values are highlighted in bold. The top three positive and negative ΔΔSvib ENCoM values are highlighted in bold.

Intramolecular interactions are altered due to mutations in N-protein

Next, we sought to closely analyse the changes in the intramolecular interactions in some of the mutants that exhibited significant alterations in ΔΔG. We compared the intramolecular interaction for T271I (ΔΔG: 1.184 kcal/mol) and I292T (ΔΔG: −1.952 kcal/mol) as these two mutants showed maximum variations among thirty eight mutants present in RBD and dimerization domain of N-protein (Tables 4 and 5). Our data clearly showed the variations in the interactions mediated by wild type and mutant residues in the pocket, where these amino acids resides as shown in Figs. 4A and 4B (T271I), and 4C and 4D (I292T). Altogether, our data strongly suggests that the mutants identified in our study are affecting the dynamic stability as well as intramolecular interactions in the N-protein.

Figure 4 Analysis of interatomic interactions.

Visual representation of interatomic interactions contributed by T271I and I292T of N-protein. Both of these mutants showed maximum positive and negative ΔΔG among mutants present in RBD and dimerization domain of N-protein. (A & B) Threonine to isoleucine substitution at 271st position; (C & D) isoleucine to threonine substitution at 292nd position. Wild-type and mutant residues are represented in light-greencolor. The interactions made by wild type and mutant residues are highlighted in each panel. The polar interactions are depicted in red dotted line, hydrophobic interaction in green and weak hydrogen bonds in orange.

Discussions

SARS-CoV-2 is an RNA virus, a causative agent of COVID-19. This virus spread worldwide within a span of few months and during its spread it also acquired mutations. Several recent studies reported the appearance of mutations in SARS-CoV-2 proteins (Korber et al., 2020; Pachetti et al., 2020; Chand, Banerjee & Azad, 2020b). This study was performed with an aim to identify mutations in N-protein which is one of the main structural proteins of SARS-CoV-2. Here, we analysed 4,163 sequences of N-protein from USA and identified 107 mutations upon comparison from first reported sequences of the same protein from Wuhan, China. We also observed around 64% (69 out of 107) of these mutations reside in the IDRs of N-protein. Among IDRs, the IDR2 harbours 36 mutations that correspond to the most number of mutations observed in a single distinct region of the N-protein.

Earlier studies demonstrated that Ser and Arg-rich linker region (IDR2) plays indispensible role in intracellular signalling events primarily by phosphorylation at Ser residues (Wootton, Rowland & Yoo, 2002; McBride, Van Zyl & Fielding, 2014). The wild type LKR/ IDR2 contains sixteen Ser residues, and our study revealed that out of those, twelve serine residues are mutated (Table 2). Therefore, we can safely assume that these mutations of Ser residues might contribute to alteration of phosphorylation dependent signalling. A recent study shows that S197, S202, R203 and G204 are important sites of phosphorylation by Aurora kinase A/B, GSK-3 as well as for its interactions with 14-3-3 protein (Tung & Limtung, 2020). Surprisingly, our study report mutation in all of these four residues suggesting that these mutant might have altered phosphorylation signaling. We have also observed that R203 and G204 is the most frequently mutated residue of N-protein (Table 6). Similar observations were also reported from other locations (Franco-Munoz et al., 2020). Furthermore, two recent independent studies revealed that SARS-CoV-2 is capable of suppressing the type-I IFN innate immune pathway possibly due to the role of N-protein in signalling events (Blanco-Melo et al., 2020; Zhou et al., 2020a) which can potentially alter the virulence of SARS-CoV-2.

We also measured ΔΔG and ΔΔSVibENCoM for the mutants that reside in the RBD and dimerization domain of N-protein. The four mutants that exhibited highest values for ΔΔG and ΔΔSVibENCoM identified in our study are T271I, I292T, G284E and P364L. Since, all of them are in the dimerization domain; therefore, it is possible that these mutations might lead to alteration in the dimerization potential of N-protein. The structural study of N-protein (C terminal domain) has revealed that residue 247–279 are essential for RNA binding (Zhou et al., 2020b) which harbours seven mutations (T247A, K249R, S250F, A252S, S255A, V270LT271I). The occurrence of these mutations in C terminal domain could possibly affect its interaction with RNA that might translate into viral RNA packaging and stability. Furthermore, the N-protein is also proposed as a candidate for vaccine development because it is known to elicit strong immunological response in SARS-CoV infected patients (Lin et al., 2003). A recent study shows that several B cell epitope of SARS-CoV were identical with SARS-CoV-2 (Ahmed, Quadeer & McKay, 2020). This study revealed that one of the most important B and T cell epitope lies between residues 305–340 of N-protein; however, our study identified multiple mutations including, P309L, M322I, S327L, T329M, T334I, D340G, D340N in that stretch. Therefore, it is possible that due to these mutations the properties of epitope might change that can affect host immunological response. Another mutation, P344S has been implicated to decrease the protein stability (Khan et al., 2020). Hence, the development of vaccines that target SARS-COV-2 N-protein must consider the mutations that occur in various populations and locations.

Evidences indicate that the N-protein of coronaviruses functions as an RNA chaperones (Zúñiga et al., 2007, 2010) and also contributes to packaging and maintenance of the RNA genome. It is also involved in RNA metabolism because N-protein interaction assays have shown the core stress granule components G3BP1 and G3BP2 are its interacting partners (Gordon et al., 2020). This interaction can either enhance stress granule induction or inhibit stress granule formation by sequestering G3BP1/G3BP2 (Hou et al., 2017). Hence, the drugs that can either inhibit the interactions of RNA with N-protein or interfere with dimerization of N-protein can be a potential antiviral candidates (Lo et al., 2013). One such drug is Nucleozin and its derivatives that targets ribonucleoprotein formation in influenza virus by interfering N-protein oligomerization (Gerritz et al., 2011). Furthermore, a recent study was conducted to identify inhibitors of SARS-CoV-2 N-protein, identified various promising candidate drugs including Conivaptan, Ergotamine, Venetoclax and Rifapentine (Onat Kadioglu, 2020). These candidate drugs interact with the residues that are either mutated (residue 154, 155, 156, 166) or are in the close vicinity of the mutations (residue 67, 81, 163, 169) identified in our study. Furthermore, bioinformatics analysis predicted Dihydroergotamine , Rifabutin and Nystatin as a potential candidate drugs (Onat Kadioglu, 2020) that interacts with a stretch of residues (from residues 150–160) of N-protein. Surprisingly, our study revealed that this stretch harbour four mutations (151, 152, 154 and 156), which can potentially alter the interactions of these drugs with N-protein. Altogether, the mutation revealed in this study can interfere with various aspects of N-protein functions that include oligomerization, interaction with RNA and interference in N-protein mediated signalling events.

Conclusions

In this study we identified 107 mutations in N-protein of SARS-CoV-2 reported from USA. Further, we demonstrate these mutations can potentially alter dynamic stability of N-protein. Altogether, the data presented here, warrants further investigations to understand its impact on SARS-CoV-2 phenotype and drugs that target N-protein.

Supplemental Information

Supplemental Information 1 List of protein accession number used in this study.

The table shows the protein accession identifier numbers of N prtoein used in this study. All of these sequences are downloaded from NCBI virus database. The list contains a total of 4163 accession numbers. All of these sequences were reported from USA.

Click here for additional data file.

Additional Information and Declarations

Competing Interests

Author Contributions

Data Availability

The author declares that they have no competing interests.

Gajendra Kumar Azad conceived and designed the experiments, performed the experiments, analyzed the data, prepared figures and/or tables, authored or reviewed drafts of the paper, and approved the final draft.

The following information was supplied regarding data availability:

Raw data is available at the NCBI-Virus-Database and sequence accession numbers are available in a Supplemental File.

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
