# Peer review of "Identification and molecular characterization of mutations in nucleocapsid phosphoprotein of SARS-CoV-2"

_PeerJ, doi:10.7717/peerj.10666_

## Round 0.1 · original submission · Minor Revisions

There are some valuable suggestions from the reviewers that I believe should be addressed to improve the quality of your manuscript. Please review these suggestions. There is consensus that the article is very much in the scope of PeerJ and is of significant interest.

Reviewer 1 ·

Basic reporting

The article is well written.
The literature should provide a more detail discussion about other mutations found in the genomic sequences of SARS-CoV-2. for example see Korber et al., 2020 and Pachetti M. et al., 2020
Article structure, figures and table are well presented.
The manuscript is not self-contained. The results are relevant to the hypothesis.

Experimental design

The manuscript is within AIMS and Scope of PeerJ.
Research questions and details are relevant and well defined.
The investigation was performed abiding to a very good technical standard.
Methods are described very well.

Validity of the findings

The findings are interesting and novel. All the data have been provided and the conclusions are well presented and clear.

Additional comments

The manuscript "Identification and molecular characterization of mutations in Nucleocapsid Phosphoprotein of SARS-CoV-2" was conducted to identify the variations present in N protein of SARS-CoV-2. the authors analysed 4163 reported sequence of N protein from United States of America (USA) and compared with first reported sequence from Wuhan, China. Our study identified 107 mutations that reside all over the N protein. Further, the authors show the high rate of mutations in intrinsically disordered regions (IDRs) of N protein. The study show 45% residues of IDR2 harbour mutations. The RNA binding domain (RBD) and dimerization domain of N protein also have mutations at key residues. The authors further measured the effect of these mutations on N protein stability and dynamicity and
our data reveals that multiple mutations can cause considerable alterations. Altogether, the data strongly suggests that N protein is one of the mutational hotspot proteins of SARS-CoV-2 that is changing rapidly and these mutations can potentially interferes with various aspects of N protein functions including its interaction with RNA, oligomerization and signalling events. Overall, he article is well written. Article structure, figures and table are well presented. The results are relevant to the hypothesis. The findings are interesting and novel. All the data have been provided and the conclusions are well presented and clear.
To provide a little more context to their findings, the authors should provide a more detailed discussion about other mutations found in the genomic sequences of SARS-CoV-2, including Korber et al., 2020 and Pachetti M. et al., 2020

·

Basic reporting

The pandemic caused by SARS-CoV-2 has steered the scientific community globally to explore for strategies to combat mortality and morbidity caused by COVID-19. To tackle the situation there is pressing need to understand the evolutionary forces that shape the fitness of involved biomolecules. Therefore, this study holds importance as it presents the identification and characterization of mutations in a structural protein namely Nucleocapsid Phosphoprotein (N protein) that interacts with SARS-CoV-2 genome and mediates its packing.
Utilizing bioinformatics approach authors identified 107 mutations in N protein, categorized them based on location and used DynaMut web server to access their effect on the stability and conformation of the protein.
Introduction highlights the importance of N protein in the viral life cycle. It will in interest of authors to also mention articles that highlight the importance of N protein as a potential therapeutic target. The specific suggestions are mentioned along with other comments.
The article is well structured, English is clear, unambiguous, and professional. There are some minor issues that needs to be corrected. The details of which are provided along with other comments.

Experimental design

Overall, the article is well within the scope and it will be of interest to the readers of the journal. The article is well organized into sections and sub-sections. However, to be comprehensive recent therapeutic needs should be included.
To increase the impact of the manuscript, phylogenetic analysis can be incorporated and the evolution of N protein can be discussed. Is it directed towards more flexibility or rigidity and its implications? How these changes shape virus fitness and its virulence?

Validity of the findings

The authors performed sequence alignment to identify mutation in N protein and then subjected them in Dynamut webserver to predict the effect on protein stability.

Authors can discuss how these changes in turn effect viral nucleic acid or related functions. Authors can also highlight how their work fulfills the knowledge gap in this area.

Additional comments

General Comment: Overall, the study is interesting and a useful addition to the area. The current study reveals mutations occurring in N protein and its impact on the protein stability, which is not presented in previous research. However, there are certain things that can be addressed to bridge the knowledge gap. A revision is recommended for improving the manuscript.

Authors show top 3 mutations that show rigidification in structure and another top 3 mutants that show gain in flexibility (Figure 2). These mutations have contrasting effects. How this will affect the function of N protein (RNA packaging or other function) needs to be discussed. What benefits the viral genome, making N protein more flexible or rigid? These effects can be discussed in regard to specific regions of N protein - whether IRD, RBD, or Dimerization domain where mutation is present.

A phylogenetic analysis could be performed with the available sequences using the initial sequence (from Wuhan, China) as root to see how the sequences have evolved over time. Are there any specific residues that are selected over time? What is the effect of these selections on N protein conformation?

Introduction: There are some important articles regarding N protein that can be referred. For instance, the abundant expression of N protein during infection and it being a highly immunogenic and a potential vaccine target.
References:
Ahmed SF, et al. Preliminary Identification of Potential Vaccine Targets for the COVID-19 Coronavirus (SARS-CoV-2) Based on SARS-CoV Immunological Studies. Viruses. 2020 Feb 25;12(3):254. doi: 10.3390/v12030254. PMID: 32106567; PMCID: PMC7150947.
Tai W, et al. Characterization of the receptor-binding domain (RBD) of 2019 novel coronavirus: implication for development of RBD protein as a viral attachment inhibitor and vaccine. Cell Mol Immunol. 2020;17(6):613-620. doi:10.1038/s41423-020-0400-4
Zeng W, et al. Biochemical characterization of SARS-CoV-2 nucleocapsid protein. Biochem Biophys Res Commun. 2020 Jun 30;527(3):618-623. doi: 10.1016/j.bbrc.2020.04.136. Epub 2020 Apr 30. PMID: 32416961; PMCID: PMC7190499.
Shang B, et al. Characterization and application of monoclonal antibodies against N protein of SARS-coronavirus. Biochem Biophys Res Commun. 2005 Oct 14;336(1):110-7. doi: 10.1016/j.bbrc.2005.08.032. PMID: 16112641; PMCID: PMC7092910.
Liu SJ, et al. Immunological characterizations of the nucleocapsid protein based SARS vaccine candidates. Vaccine. 2006 Apr 12;24(16):3100-8. doi: 10.1016/j.vaccine.2006.01.058. Epub 2006 Feb 8. PMID: 16494977; PMCID: PMC7115648.

Page 3, Line 76-77: It will be appropriate to refer these recent studies as well. It discusses about N protein contribution in the formation of helical ribonucleoproteins during the packaging of the RNA genome, regulation of viral RNA synthesis during replication and transcription and modulation of metabolism in infected individuals.
Cong Y, et al. Nucleocapsid Protein Recruitment to Replication-Transcription Complexes Plays a Crucial Role in Coronaviral Life Cycle. J Virol. 2020 Jan 31;94(4):e01925-19. doi: 10.1128/JVI.01925-19. PMID: 31776274; PMCID: PMC6997762.
Astuti I, Ysrafil. Severe Acute Respiratory Syndrome Coronavirus 2 (SARS-CoV-2): An overview of viral structure and host response. Diabetes Metab Syndr. 2020 Jul-Aug;14(4):407-412. doi: 10.1016/j.dsx.2020.04.020. Epub 2020 Apr 18. PMID: 32335367; PMCID: PMC7165108.

Some additional minor changes that will improve the manuscript are as as follows:
Page 3, Line 80: An appropriate reference for the statement is required.
Page 3, Line 81-86: To make text consistent either use N- and C-terminus or N and C terminus. Also, keep it uniform throughout the manuscript.
Page 3, Line 82: In the sentence “The domain present towards the N- terminus also known and RNA binding domain...”, replace it with “The domain present towards the N-terminus is also known as RNA binding domain…”.
Page 3, Line 95: Remove extra spacing “NCBI-virus- database”
Page 3, Line 97: In the sentence, “…..sequences of N protein are being deposited from USA.” It will appropriate to change it to “…..sequences of N protein were deposited from USA.”
Page 4, Line 103, 113, 131, 166: “programe” needs to be changed to “program”.
Page 4, Line 114: Please include spacing in between 6VYO and the reference.
Page 4, Line 123, 124, 135, 145, 150, 156, 157, 159: Please be consistent in using either N protein or N-protein. Also, keep it uniform throughout the text.
Page 6, Line 177-178: The text is redundant as it is already mentioned in page 4, line 117-118.

Discussion
Page 6, Line 199-211: The text presents almost the same information as presented in the introduction. It will better if this portion is removed entirely. Also, abrupt bringing of spike glycoprotein in picture might confuse the reader. Instead, authors can directly discuss their findings, the knowledge gained and its future application.
Page 7, Line 215: It will be appropriate to change “primary” to “primarily”.
Page 7, Line 233-236: It will be more informative if authors can mention specific residues of N protein to which these drugs interact and what mutations are present in those positions that author have identified.

Figures and Table Legends
Page 8: Line 250-253: The text looks redundant and needs rephrasing, as the same information is already described in page 3, line 81-86 and page 4, line 125-129.
Suggestion: The N protein comprising of 419 residues is shown. The RNA binding domain, dimerization domain, intrinsically disordered regions including IRD1, IRD2, and IRD3 are labelled.

Figure 2: It will more appealing to the readers if the panel labeling is done inside, instead of outside. It is suggested to move mutant name inside the panels. Also, author can label N- and C-terminus in figure panels. Also, keep all structures in the same orientation - Panel A seems to be in different orientation as compared to others.

Figure 3: Similarly, to Figure 2, it is suggested to move mutant name inside the panels. Also, measure atomic distances and label in corresponding panels. Legend should clearly mention what kind of interaction different dots indicate. For instance, the polar interactions are depicted in red dotted line, hydrophobic interaction in green. Only show important interactions and corresponding distances. It can be done by downloading PyMOL sessions instead of direct images from DynaMut web server. The interatomic distances can be measured in PyMOL.

Page 8, Line 270-273: Table 1: The following sentences seem redundant as author has already discussed it earlier in the text, “These mutations are present in the N protein of SARS-CoV-2 reported from USA. The first reported sequence of N protein from Wuhan, China was used as wild type sequence for this analysis. The table shows only those residues that have variation, rest of the sequences are identical among all samples.”. Suggestion is to remove it or rephrase.

The same text is repeated from Table 1 legend. Removal of these sentences is recommended in the following:
Page 9, Line 275-279: Table 2
Page 9, Line 282-285: Table 3
Page 9, Line 288-291: Table 4
Page 9, Line 294-297: Table 5

Another minor suggestion is to use consistent scheme for tables, either use smaller or capital letter in the header. In some places author has mentioned wild type residue while in other it is Wild type residue. Also, S.No in table 4 is S.No. in other tables.

Reviewer 3 ·

Basic reporting

'no comment'

Experimental design

'no comment'

Validity of the findings

'no comment'

Additional comments

In this work, Azad Gajendra K performed sequence analysis of 4163 sequences of N protein of SARS-CoV-2 from United States of America to identify the variations present in it. This study identified 107 mutations those are distributed all over the N protein. The analysis also finds mutations in RNA binding domain (RBD) and dimerization domain of N protein. The effects of these mutations were predicted using DynaMut programe and concluded that some mutations could possibly hamper N protein functions. Overall, this is reasonable analysis that would advance our understanding of variations in SARS-CoV-2 genome. However, some changes could further improve the quality of the manuscript.

1) What is probability of mutations- representing it in logo plot would be helpful.
2) Show mutations (structural representation) in carton and surface diagram of crystal structure and highlighting mutated residues in one color for overall distribution of mutations in the protein.
The structure representation is not properly presented. It should be like that, for example a figure ligand be like this:
“Cartoon and surface representation of crystal structure of the ---- with mutations shown in red spheres (PDB ID:…..). All structural representations are made using …… software (PyMOL ? https://www.pymol.org)

3) Figure 3 is really hard to interpret. Show side chain as a stick representation without any hydrogens (in structural figure generating program do- hide hydrogens-all) for better and clear figure generation. These current figures are outcome from DynaMut programe but should be generated using program PyMOL which is simple and can generate descriptive figures.

---

## Round 0.2 · Minor Revisions

Thank you for your submission and response to the reviewers' comments.

Please see the remaining comments from reviewer 3 and either revise the manuscript in line with them or provide a rebuttal letter explaining why you prefer not to.

·

Basic reporting

Well presented

Experimental design

Well defined

Validity of the findings

Novel and clear

Additional comments

The author has revised the manuscript carefully and resolved all the raised concerns. The work is appreciable. The revised manuscript in the current form seems suitable for publication. It will be of interest to the readers of the journal.

Reviewer 3 ·

Basic reporting

no comment

Experimental design

no comment

Validity of the findings

no comment

Additional comments

In this revision, Author could have added reviewers suggested changes. I have suggested some structural figures changes but ignored. Also, I suggested to add weblogo diagram and similarly one reviewer suggest to add phylogenetic analysis but that is also ignored in the revision. This changes could immensely improve the quality of the manuscript.

---

## Round 0.3 · Minor Revisions

Please address the couple of comments made by Reviewer 2.

·

Basic reporting

No comment

Experimental design

No comment

Validity of the findings

No comment

Additional comments

In the current version, the incorporation of weblogo is a valuable addition. However, there are concerns regarding the logo and some suggestions for improvement of manuscript.
1) Page 5, Line 138: The author needs to describe how the weblogo was generated. If referring to another study, briefly describe the procedure that author followed providing the details how many sequences used. Author should provide the multiple sequence alignment file or the set of N-protein sequences that was used to generate weblogo as supplementary data.

2) Figure 1, Panel D: The presented weblogo needs to be updated. Is it generated from multiple sequence alignment of N-protein? The current weblogo is not representing the relative frequency of each amino at that position.
In the present context, the weblogo should basically be a graphical representation of an amino acid multiple sequence alignment of N-protein. As author has mentioned in the figure legend the overall height of the stack indicates the sequence conservation at that position, while the height of symbols within the stack indicates the relative frequency of each amino at that position. But relative frequency is not observed.

3) Figure 1, Panel B and C: There is a suggestion to improve the quality of these figures. The author can download UCSF Chimera, which is free for academic research. It is easy to use and provides high resolution images. Author can render these structures in Chimera, use white background as in other figures and label residues shown in sticks. This will improve the clarity and quality. Also, mention the PDB ID in legend.

4) Figure 3 and 4 panels: The DynaMut webserver provides either PyMol sessions or high-resolution images of predictions. The analysis of prediction using the PyMol session file and mapping the interatomic distances would have useful if author had license to PyMol. Although, the free source code is available but requires compilation of libraries and can be difficult. Author can try Pre-compiled Open-Source PyMOL, if well versed with command line.
https://pymolwiki.org/index.php/Windows_Install
https://www.lfd.uci.edu/~gohlke/pythonlibs/#pymol-open-source
Author seems to have used image files of prediction because of the limitations.

Reviewer 3 ·

Basic reporting

no comment

Experimental design

no comment

Validity of the findings

no comment

Additional comments

Author have done some suggested changes, Accept the manuscipt. However this is strange that author is not able to download and follow instructions on how to use freely available software to scientific community.

---

## Round 0.4 · accepted · Accept

I am happy to accept the current revision, given that the critical changes have been made, and since Reviewer 3 accepted Version 2, and Reviewer 2 accepted Version 1.